# Sarcopenia and Ghrelin System in the Clinical Outcome and Prognosis of Gastroenteropancreatic Neuroendocrine Neoplasms

**DOI:** 10.3390/cancers14010111

**Published:** 2021-12-27

**Authors:** Yiraldine Herrera-Martínez, Carlos Alzas Teomiro, Soraya León Idougourram, María José Molina Puertas, Alfonso Calañas Continente, Raquel Serrano Blanch, Justo P. Castaño, María Ángeles Gálvez Moreno, Manuel D. Gahete, Raúl M. Luque, Aura D. Herrera-Martínez

**Affiliations:** 1Department of Nuclear Medicine, Virgen del Rocio University Hospital, 41013 Seville, Spain; gerah10@gmail.com; 2Maimonides Institute for Biomedical Research of Cordoba (IMIBIC), Reina Sofia University Hospital, 14004 Cordova, Spain; carlosalzas@hotmail.com (C.A.T.); sorayaleon@gmail.com (S.L.I.); cmmolina@hotmail.com (M.J.M.P.); acalanas@hotmail.com (A.C.C.); rserrano@hotmail.com (R.S.B.); justocastano@gmail.com (J.P.C.); mariaagalvez@juntadeandalcia.es (M.Á.G.M.); bc2gaorm@uco.es (M.D.G.); bc2luhur@uco.es (R.M.L.); 3Endocrinology and Nutrition Service, Reina Sofia University Hospital, 14004 Cordova, Spain; 4Medical Oncology Service, Reina Sofia University Hospital, 14004 Cordova, Spain; 5Department of Cell Biology, Physiology, and Immunology, University of Córdoba, 14014 Cordova, Spain; 6CIBER Fisiopatología de la Obesidad y Nutrición (CIBERobn), Instituto de Salud Carlos III, 14004 Cordova, Spain

**Keywords:** NENs, sarcopenia, CT scan, ghrelin system, nutrition, outcome

## Abstract

**Simple Summary:**

Malnutrition and sarcopenia affect clinical outcomes in cancer patients. Nutritional evaluation in patients with neuroendocrine neoplasms (NENs) is not routinely performed. Currently, the evaluation of sarcopenia using CT scans is the gold standard in cancer patients, additionally, anthropometric, biochemical and molecular analysis of patients with gastroenteropancreatic NENs at diagnosis was perfomed. The expression levels of key ghrelin system components were assessed in 63 tumor samples. Results: Nutritional parameters were similar in GEP-NEN tumors of different origin. Relapsed disease was associated with decreased BMI. Patients who presented with weight loss at diagnosis had significantly lower overall survival (108 (25–302) vs. 263 (79–136) months). Ghrelin O-acyltransferase (GOAT) enzyme expression was higher in these patients. The prevalence of sarcopenia using CT images reached 87.2%. Mortality was observed only in patients with sarcopenia. Muscle evaluation was correlated with biochemical parameters but not with the expression of ghrelin system components. Conclusion: Survival is related to the nutritional status of patients with GEP-NENs and also to the molecular expression of some relevant ghrelin system components. Routine nutritional evaluation should be performed in these patients, in order to prescribe appropriate nutritional support, when necessary, for increasing quality of life and improving clinical outcomes.

**Abstract:**

Background: Malnutrition and sarcopenia affect clinical outcomes and treatment response in cancer patients. Patients with neuroendocrine neoplasms (NENs) may present with additional symptoms related to tumor localization in the gastrointestinal tract and hormone secretion, increasing the risk and effects of sarcopenia. Aim: To explore the presence of malnutrition and sarcopenia in gastroenteropancreatic (GEP)-NEN patients, their relation to tumor characteristics, patient outcomes, survival and the molecular expression of ghrelin system components in the tumor. Patients and methods: One-hundred-and-four patients were included. Anthropometric, biochemical and CT-scans at diagnosis were evaluated. The expression levels of key ghrelin system components were assessed in 63 tumor samples. Results: Nutritional parameters were similar in GEP-NEN tumors of different origin. Relapsed disease was associated with decreased BMI. Patients who presented with weight loss at diagnosis had significantly lower overall survival (108 (25–302) vs. 263 (79–136) months). Ghrelin O-acyltransferase (GOAT) enzyme expression was higher in these patients. The prevalence of sarcopenia using CT images reached 87.2%. Mortality was observed only in patients with sarcopenia. Muscle evaluation was correlated with biochemical parameters but not with the expression of ghrelin system components. Conclusion: Survival is related to the nutritional status of patients with GEP-NENs and also to the molecular expression of some relevant ghrelin system components. Routine nutritional evaluation should be performed in these patients, in order to prescribe appropriate nutritional support, when necessary, for increasing quality of life and improving clinical outcomes.

## 1. Introduction

Malnutrition is common in cancer patients, which can be a result of the tumor itself but also of the medical or surgical treatment. According to the European Society of Parenteral and Enteral Nutrition (ESPEN), 10–20% of cancer patients die due to consequences of malnutrition rather than the tumor itself, since malnutrition affects treatment tolerance and response [1]. In this context, an appropriate nutritional screening should be performed at diagnosis and during follow-up in order to establish appropriate therapeutic nutritional strategies [2]. 

Currently, nutritional interventions cannot be based on the evaluation of body mass index (BMI) due to the increased prevalence of overweight and obesity. Additionally, visceral obesity, sarcopenia and sarcopenic obesity have been identified as adverse factors in cancer patients. Furthermore, the excess of body fat has been also associated with several types of cancer [3,4]. For these reasons, specific tools have been designed for obtaining an initial screening for malnutrition in cancer patients [5]. When screening is positive, an appropriate nutritional evaluation should be performed in order to determine the presence of sarcopenia and start specific nutritional treatment [1].

Sarcopenia is defined as a loss of muscle mass, strength and physical performance, and it is associated with altered amino acid metabolism, increased muscle protein catabolism relative to anabolism, and loss of muscle fibers [6]. Sarcopenia evaluation requires specific techniques that are not regularly performed. Currently, computed tomography (CT) and magnetic resonance imaging (MRI) are considered the gold standards for estimating muscle mass in cancer patients [7], but imaging evaluation of sarcopenia is not routinely performed.

Additionally, in patients with neuroendocrine neoplasms (NENs), nutritional status can be also affected by hormone secretion, which can produce malabsorption, diarrhea, steatorrhea, altered gastrointestinal motility and weight loss, among other symptoms [3]. Despite this, NENs are indolent tumors in several cases. Treatment is usually well-tolerated, and patients present with high survival rates (median survival duration of 41 months) [8]. In consequence, nutritional interventions are not routinely performed in these patients, and studies that specifically evaluate the nutritional status of patients with NENs are limited [9].

The ghrelin system is involved in the regulation of multiple physiological functions, including hormonal secretion, regulation of appetite, gastric motility, body composition and reduced energy expenditure [10,11,12,13,14]. Ghrelin hormone must undergo a unique modification, consisting of the acylation of the third serine residue, which is catalyzed by the ghrelin-O-acyl-transferase (GOAT) enzyme [13,15]. This acylated ghrelin represents the active form that binds the canonical ghrelin receptor, GHSR1a, but there is also a truncated receptor GHSR1b, with an unknown ligand and function [10,16,17]. Ghrelin is a multifunctional hormone and its expression has been demonstrated in several tissues, especially in glands and in the gastrointestinal tract. Ghrelin plays a role in the regulation of energy balance, gastric acid release, appetite, insulin secretion, gastric motility and the turnover of gastric and intestinal mucosa [18]. In this context, the ghrelin system might play an important role in the regulation of cancer-related processes [19,20,21,22] and antagonizing protein breakdown in the catabolic conditions of cancer cachexia [23]. Our group has described its presence in GEP-NENs and some clinical relations with tumor progression [24]. Despite this, its specific role in body composition and/or malnutrition in NEN patients is still unknown.

In this context, this study was performed in order to evaluate the nutritional status, and presence of sarcopenia, in GEP-NEN patients at diagnosis (using anthropometric, biochemical and imaging techniques), its association with tumor characteristics and its influence on patient survival. Additionally, we aimed to explore some putative relations between nutritional status, sarcopenia and the molecular expression of key components of the ghrelin system in tumor samples, in order to improve the clinical diagnosis of malnutrition and sarcopenia in NEN patients.

## 2. Material and Methods

### 2.1. Patients

This study was conducted in accordance with the Declaration of Helsinki and the Ethics Committee. Patients were treated following national and international clinical practice guidelines [25,26,27]. Written informed consent was required before inclusion. Data from 104 patients with GEP-NENs were collected (demographic and clinical characteristics of the cohort are summarized in Table 1). Patients were diagnosed between 2001–2014 in a single hospital. Additionally, formalin-fixed paraffin-embedded (FFPE) tumor samples were available for analyses in 63 patients. Patients with hereditary endocrine syndrome were excluded. Clinical records were used to collect full medical history. 

### 2.2. CT Image Analysis

CT images at diagnosis were used for this analysis. Two adjacent axial images within the same series, at the third lumbar vertebra, were selected for the analysis of total muscle cross-sectional area (cm^2^) and averaged for each patient. Images were analyzed using a dedicated workstation (Carestream Vue PACS v12.0.0.0700; Carestream Health, Inc., Rochester, NY, USA) that enables specific tissue demarcation using previously reported Hounsfield unit (HU) thresholds. Skeletal muscle tissue was separated according to different density thresholds: a density value of +35 HU was used to separate fat from muscle tissue and +150 HU to separate muscle from bone tissue [28]. Our observer was trained to correctly identify and quantify lumbar vertebrae and the following muscles: rectus abdominus, abdominal (lateral and oblique), psoas, and paraspinal (quadratus lumborum, erector spinae). The observer was blinded to patient survival status. Muscle area was normalized for height in meters squared (m^2^) and reported as lumbar skeletal muscle index (SMI, cm^2^/m^2^). We used specific cut-offs for sarcopenia (L3 SMI: 38.5 cm^2^/m^2^ for women and 52.4 cm^2^/m^2^ for men) as previously described in a CT-based sarcopenic study for patients with solid malignancies [29].

### 2.3. RNA Isolation and Reverse-Transcription 

Total RNA from FFPE samples (n = 63) were isolated using the RNeasy-FFPE Kit (Qiagen, Limburg, The Netherlands) according to the manufacturer’s instructions, as previously described [24,30]. Quantification of the recovered RNA was assessed using the NanoDrop2000 spectrophotometer (Thermo Fisher Scientific, Wilmington, NC, USA). Total RNA was retrotranscribed to cDNA with the First-Strand Synthesis kit using random hexamer primers (Thermo Fisher Scientific), as previously reported [19,20,31,32,33]. 

### 2.4. Quantitative Real-Time PCR (qPCR)

cDNAs were amplified with the Brilliant III SYBR-Green Master Mix (Thermo Fisher Scientific) using the Stratagene Mx3000p system and specific primers for each transcript of interest. Specifically, expression levels (absolute mRNA copy number/50 ng of sample) of ghrelin, In1-ghrelin, GOAT-enzyme, GHSR1a and GHSR1b were measured using previously validated primers [34,35,36]. RNA expression was adjusted by *18S* gene expression [31]. This housekeeping gene was chosen after comparing its expression with *BACT* by the GeNorm 3.3 software [37]. According to the analysis, *18S* was the most reliable housekeeping gene for these samples.

### 2.5. Statistical Analysis

Between-group comparisons were analyzed by the Mann–Whitney U test (nonparametric data) or the Kruskal–Wallis test (nonparametric data, when we compared more than two groups). Paired analysis was performed with the Student’s *t*-test (parametric data) or the Wilcoxon test (nonparametric data). A chi-squared test was used to compare categorical data. Statistical analyses were performed using SPSS statistical software version 20 and GraphPad Prism version 8. Data are expressed as median ± interquartile range and percentages. *p*-values < 0.05 were considered statistically significant. 

## 3. Results

### 3.1. Patient Population and Clinical Evolution

One-hundred-and-four patients were included. Most of them were female (54.8%) with a median age of 54 years at diagnosis and presented mainly with pancreas NENs (38.5%). Most patients presented with grade 1 tumors (33.7%); patients with neuroendocrine carcinoma were not included. Almost 40% of patients presented with weight loss at diagnosis despite a median BMI of 27.2 Kg/m^2^ (Table 1). 

At diagnosis, most patients presented with comparable biochemical levels of serum proteins (transferrin, albumin, ferritin, prealbumin and reactive C protein (RCP)), lipids (total and fractionated cholesterol) and lymphocytes, considering the primary tumor site (Figure 1A). Additionally, the presence of metastasis at diagnosis (almost 50% of the included patients) tended to be associated with lower BMI and serum LDL cholesterol levels. Furthermore, patients with relapsed disease presented lower BMI whereas patients who were disease-free presented with higher BMI at diagnosis. Serum albumin levels also tended to be associated with increased mortality in NEN patients (Figure 1B).

Weight loss at diagnosis was more prevalent in patients with grade 2 tumors, as well as decreased total cholesterol levels and increased RCP (Appendix A). These results were not statistically significant.

### 3.2. Cumulative Survival in NEN Patients Is Associated with Nutritional Parameters 

Patients who presented at diagnosis with weight loss had significantly lower overall survival compared with patients who had not presented this symptom (108 (25–302) vs. 263 (79–136) months) (Figure 2A; Appendix A). A similar effect on survival was observed when the presence of metastasis at diagnosis was evaluated (136 vs. 245 months; *p* = 0.05; Figure 2B). Additionally, patients with decreased albumin at diagnosis tended to present lower overall survival compared with patients with normal serum levels (65 vs. 142 months; *p* = 0.06; Figure 2C). Specific data about Log Rank and IQR are presented in Appendix A. BMI and other biochemical parameters (transferrin, ferritin, lymphocytes, RCP and cholesterol) were not associated with survival in NEN patients (Appendix A). 

### 3.3. The Molecular Expression of the Ghrelin System and Its Relation to Nutritional Characteristics of NEN Patients

Since weight loss and serum albumin were the most relevant clinical symptoms/biochemical markers associated with survival in NEN patients, we further evaluated their relation to the molecular expression of ghrelin system components in tumor samples of 63 available patients. The expression levels of the GOAT enzyme were significantly increased in patients who presented with weight loss at diagnosis (Figure 3A). Other system components, including ghrelin hormone and the splicing variants In1-ghrelin and GHSR1b, were also increased, but results were not statistically significant. Regarding serum albumin levels, no significant relation was observed between decreased albumin and the molecular expression of ghrelin system components (Figure 3B).

Additionally, ghrelin expression levels were correlated with weight at diagnosis, while the native and truncated receptors (GHSR1a and GHSR1b, respectively) were associated with the number of survival months. Finally, GHSR1b was also significantly correlated with some biochemical nutritional parameters, including ferritin, total and LDL cholesterol (Figure 3C). Finally, all ghrelin system components were compared using ROC curves for evaluating weight loss in NEN patients, wherein GOAT was the most accurate marker with an AUC of 0.755 (Figure 3D). 

### 3.4. Sarcopenia at Diagnosis Found to Be Highly Prevalent in NEN Patients Using CT Scan Analysis 

A prevalence of sarcopenia at diagnosis of 87.2% was observed using CT scan imaging. Serum parameters were significantly correlated with paravertebral, abdominal and psoas areas, as well as with SMI (Figure 4A). Any specific clinical association between muscle areas, densities and SMI was obtained. Mortality occurred only in patients without sarcopenia (Figure 4B), despite the fact that results were not statistically significant. Finally, the expression levels of ghrelin system components were similar in patients with and without sarcopenia defined by decreased SMI (Figure 4C). Decreased SMI was also more prevalent in patients with grade 2 tumors (Appendix A). No relation between sarcopenia and primary tumor site was observed in our cohort (results not shown).

## 4. Discussion

This study aimed to evaluate the nutritional status at diagnosis of patients with GEP-NENs using epidemiological, anthropometrical, biochemical and imaging techniques, and to determine putative relations with the molecular expression of key components of the ghrelin system. There are some studies focused on nutritional aspects in NEN patients [38,39]; however, to our knowledge, this is the first study that comprehensively characterizes the nutritional status and the prevalence of sarcopenia using CT scan imaging for a representative cohort of 104 well-differentiated GEP-NENs. Moreover, we analyzed the impact of nutritional status at diagnosis for disease progression and survival after 7–20 years of follow-up. Furthermore, although some studies have reported the presence of certain components of the ghrelin system in GEP-NENs [21,24,40,41,42], to the best of our knowledge, its relation to nutritional parameters has not been specifically described in these tumors. Overall, our results revealed that weight loss at diagnosis predicts lower survival in NEN patients. The prevalence of sarcopenia reaches 87% of patients in our cohort, suggesting that the diagnosis of sarcopenia using CT scans might help to detect these patients early on. Finally, some key components of the ghrelin system, especially the GOAT enzyme, displayed alterations and clinical associations related to the nutritional status of GEP-NEN patients.

Importantly, the American Society for Parenteral and Enteral Nutrition recognized albumin and prealbumin as important factors that correlate with the risk for adverse outcomes in patients, though they should not be used for diagnosing protein-energy malnutrition [43]. This statement is based on the fact that there is an association between inflammation and malnutrition, but not between malnutrition and visceral protein levels [43]. In our cohort we observed that albumin tended to correlate with survival in NEN patients, but this fact suggests that the evaluation of serum levels of visceral proteins is not enough for the diagnosis of malnutrition in cancer patients. Furthermore, serum albumin was correlated with muscle area and SMI using CT images, an observation that has been previously described [44], suggesting the importance of its determination in these patients in combination with other techniques for nutritional evaluation.

Weight loss is considered an important factor for diagnosing and classifying malnutrition [45]. Indeed, it has been associated with increased mortality in solid tumors, independently of age, race and tumor stage [46,47,48]. This study demonstrates decreased survival in NEN patients who presented with weight loss at diagnosis. Related to this, an observational study performed with 203 NEN patients reported increased length of hospital stays and decreased overall survival in patients with malnutrition assessed by screening tools, anthropometric and biochemical variables [49].

Despite malnutrition having been widely reported as a prevalent factor in cancer patients, there is limited data regarding malnutrition prevalence in NEN patients, especially its relation to clinical outcomes, survival and prognosis [38]. Recent publications suggest that at least 21–25% of NEN patients might present with malnutrition [49,50] such that regular nutritional screening and evaluation in these patients at diagnosis would be appropriate [9]. Moreover, current clinical guidelines suggest the relevance of evaluating muscle mass loss (in combination with BMI, weight loss, food intake and inflammatory status) for diagnosing malnutrition [45], which highlights the necessity of using specific techniques for evaluating muscle mass in cancer patients. Specifically, the use of low skeletal muscle mass as the definition for sarcopenia may help clinicians to make treatment decisions more conveniently and more quickly [51], since BMI does not discriminate the prevalence of sarcopenia or myosteatosis [52].

In this context, it is necessary to use additional tools for evaluating malnutrition in cancer patients, especially in patients with NENs. Since the determination of sarcopenia using CT scans represents the gold standard for evaluating body composition in cancer patients [7], it seems reasonable to evaluate this parameter since most patients have a CT or MRI scan that includes the paravertebral area at diagnosis and during follow-up. Recently, a study performed with 49 metastatic NENs that underwent peptide receptor radionuclide therapy (especially pancreatic and small bowel NENs) reported a prevalence of 67% sarcopenia and 71% myosteatosis using CT scans. In this study, progression-free survival was similar in patients with or without sarcopenia, and 12% of patients died during the follow-up [38]. In contrast, in our cohort, the mortality rate was higher and the prevalence of sarcopenia reached 87%. Importantly, we did not analyze the specific effect of any treatment. In line with this, another retrospective study performed in gastric NENs reported a prevalence of sarcopenia (using CT scans) of 42.8% [39]. In this study, a higher incidence of sarcopenia in the subgroups of male patients, aged 65 years, with a BMI of <25 Kg/m^2^ and a tumor larger than 50 mm was observed [39]. Importantly this study included only patients with gastric neuroendocrine tumors, neuroendocrine carcinomas and gastric mixed adenoneuroendocrine carcinomas. The authors describe that sarcopenia was not associated with short-term clinical outcomes of these patients (total postoperative complications, surgical complications and systemic complications) but it was an independent risk factor for long-term complications in patients with gastric mixed adenoneuroendocrine carcinomas [39]. Results may differ from our results since we did not evaluate post-surgical complications. Additionally, the number of well-differentiated gastric NENs was limited; thus, the prognostic value of sarcopenia for gastric NENs may be biased. Additionally, this study was conducted in an Asian population with only 23.9% females [39]. 

Based on this, it is reasonable to suggest that appropriate nutritional screening should be performed at diagnosis and during follow-up for all patients with NENs, as in other types of cancer [1]. Additionally, nutritional recommendations, including calorie and protein intake should be provided [53], and additional oral nutritional supplementation should be started early on for those patients who are malnourished or at risk of malnutrition [1].

In relation to this, ghrelin is an orexigenic gut hormone secreted by several tissues which transmits hunger signals from the peripheral to the central nervous system [54]. It has the potential to increase body weight and body composition through increased appetite, growth hormone secretion and gastric acid secretion. Ghrelin also prevents muscle catabolism, promotes gut motility and regulates metabolism [13,55,56]. These diverse actions suggest that it may disturb the vicious cycle of cachexia through its anabolic, orexigenic and anti-inflammatory effects [13]. There is increasing evidence that suggests that the ghrelin system might be involved in regulating several processes related to cancer progression, especially in metastasis and proliferation [18]. Ghrelin, its native receptor (GHSR1a) and the truncated receptor (GHSR1b), have been described in renal cell carcinomas and neuroendocrine tumors, including pituitary, pancreatic, thyroid, lung, breast, gonadal, prostate, ovarian, oral, gastric and colorectal cancer, using inmunohistochemistry or RT-PCR [42,57,58]. Additionally, some studies have demonstrated correlations between the expression of this system, tumor progression and survival outcome in patients with cancer, including renal carcinomas and GEP-NENs [24,58,59]. Ghrelin may also influence cell migration and invasion capacity and consequently metastasis in several types of cancer by the GHSR/PI3K/Akt signaling pathway; in contrast, its role in breast and prostate cancers is controversial [18,60,61].

In this context, in NENs, our and other groups have suggested a role for the ghrelin system in the regulation of GEP-NEN pathophysiology [11,12,24,40]. Despite this, to the best of our knowledge, specific studies reporting the expression of ghrelin system components and its nutritional relevance in NEN have not been published yet. Our group previously described a marked overexpression of GOAT in GEP-NEN tumors, which was associated with increased tumor size [24]. Interestingly, in our current study, this ghrelin system component was increased in patients who presented weight loss at diagnosis and consequently decreased survival. Importantly, we did not observe significant changes between the molecular expression of ghrelin system components in patients with or without sarcopenia using CT scans, but these results might be related to the sample size.

This study has some limitation: it is a retrospective cohort study, in which a specific nutritional anthropometric evaluation was not performed, as currently suggested [62]. In consequence, some data, including bioimpedance analysis, are not available. Additionally, due to the number of patients, the effect of specific therapies was not evaluated, and the circulating levels of ghrelin components were not available. Despite this, several strengths of this study should be highlighted, since it includes a well-characterized cohort of well-differentiated NENs of different origin, with similar proportions of males and females and with sarcopenia evaluation using CT scans, which represent the current gold standard for evaluating body composition in cancer patients [7]. This study also includes patients who were followed-up over a long period of time.

## 5. Conclusions

In summary, we present a characterization of the nutritional status of GEP-NEN patients at diagnosis, the prevalence of CT-based sarcopenia, its relation to survival and the molecular expression of key components of the ghrelin system. Our results demonstrate that sarcopenia is highly prevalent in NEN patients, suggesting the necessity of appropriate nutritional screening/evaluation at diagnosis to identify patients with putatively worse prognoses and to establish specific therapeutic options for reversing malnutrition and improving quality of life and clinical outcomes. Prospective studies that include larger cohorts are still necessary to add to and validate the information provided in this study; circulating levels of ghrelin system components might also provide additional information, since these molecular targets, especially GOAT, could represent putative prognostic markers for GEP-NENs.

## Figures and Tables

**Figure 1 cancers-14-00111-f001:**
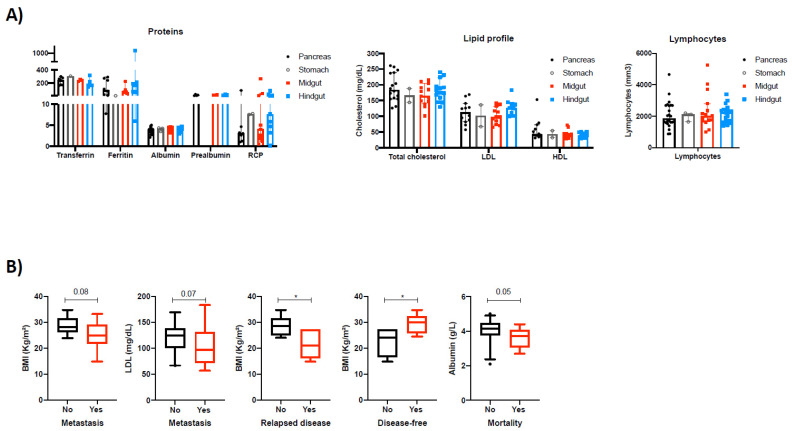
(**A**) Biochemical parameters for assessing nutritional status in patients with NENs according to the primary tumor site. (**B**) clinical associations between nutritional parameters and clinical outcome in NEN patients. *: *p* < 0.05.

**Figure 2 cancers-14-00111-f002:**
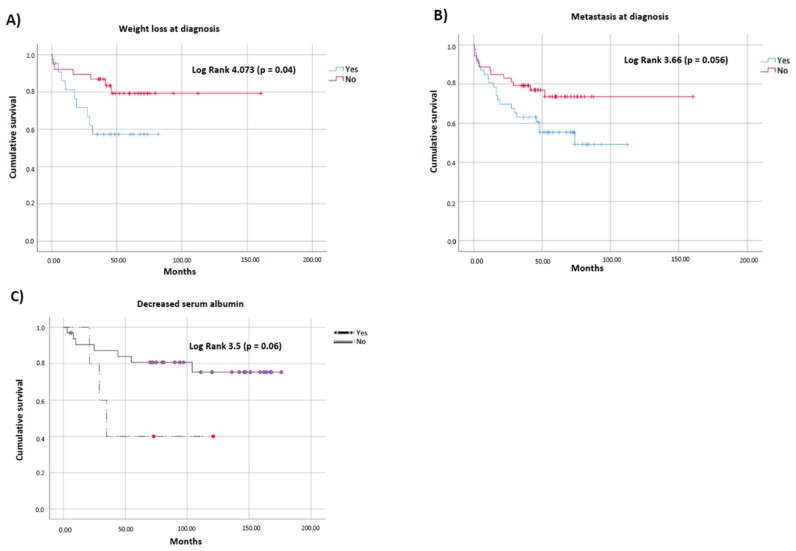
Cumulative survival curves for NEN patients according to the presence of nutrition-related parameters at diagnosis: (**A**) weight loss; (**B**) metastasis; (**C**) decreased serum albumin levels.

**Figure 3 cancers-14-00111-f003:**
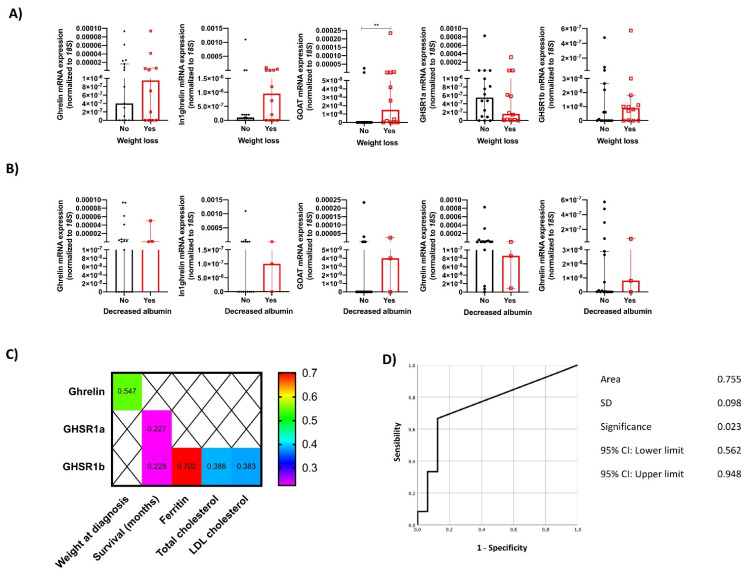
Molecular expression of ghrelin system components in NEN tumor samples according to the presence of: (**A**) weight loss; (**B**) decreased serum albumin levels; (**C**) significant clinical correlations between the mRNA expression of some ghrelin system components and nutrition-related parameters in NENs; (**D**) ROC curve representing GOAT enzyme as marker for identifying NEN patients with weight loss. **: *p* < 0.01.

**Figure 4 cancers-14-00111-f004:**
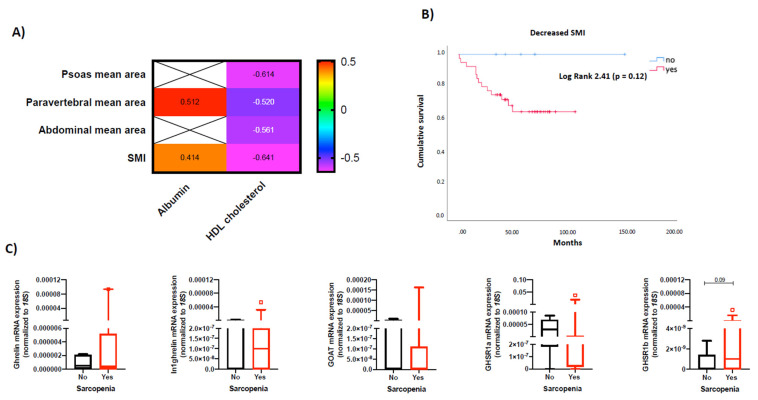
(**A**) Significant clinical/biochemical correlations between nutritional parameters and CT-measured muscle characteristics including SMI. (**B**) Cumulative survival curve for NEN patients according to the presence of sarcopenia at diagnosis using CT imaging. (**C**) Molecular expression of ghrelin system components in NEN tumor samples according to the presence of sarcopenia at diagnosis using CT scans.

**Table 1 cancers-14-00111-t001:** Baseline clinical and biochemical characteristics.

Characteristic	Total (*n* = 104)
Sex (♂/♀)	45.2/54.8
Age at diagnosis (years)	54.5 (52–58)
Functional tumors	32.9 (27/82)
Incidental tumor	38.7 (29/75)
Tobacco exposure	
No	31.1 (14/45)
Active	48.9 (22/45)
Previous exposure	20 (9/45)
Other neoplasms	20 (18/90)
Tumor localization	
Pancreas	38.5 (40/104)
Stomach	4.8 (5/104)
Small bowel	21.2 (22/104)
Hindgut	34 (35/104)
Other	2 (2/104)
Nutritional characteristics	
Weight loss at diagnosis	39.7 (25/63)
Weight at diagnosis (Kg)	70 (65– 78)
BMI at diagnosis	27.2 (24.5–28.7)
Metastasis at diagnosis	49.1 (51/104)
Liver	17.4 (8/46)
Spleen	2.2 (1/46)
Lymph nodes	50 (23/46)
Peritoneum	2.2 (2/46)
Multiple invasion	28.2 (13/46)
Relapsed disease	25.4 (17/67)
Disease-free	75.9 (44/58)
Mortality	34.6 (36/104)
Histologic features	
Necrosis	7.3 (9/33)
Multiple tumors	7.5 (4/53)
Peritumoral invasion	51.8 (44/85)
Vascular invasion	28.2 (22/78)
Perineural invasion	28 (21/75)
Tumor grade	
Grade 1	33.7 (35/104)
Grade 2	26.9 (28/104)
Grade 3	9.6 (10/104)
Unknown	29.8% (31/104)
Biochemical analysis at diagnosis	
Lymphocytes	1520 (390–2762)
Transferrin (mg/dL)	240 (199–262)
Ferritin (mg/dL)	81.7 (5–63)
Albumin (g/dL)	3.9 (1.6–5.6)
Prealbumin (mg/dL)	23 (9.7–41.6)
RCP (g/dL)	2.6 (1.3–5.4)
Total cholesterol (mg/dL)	165 (30–206)
LDL cholesterol (mg/dL)	95 (53–124)
HDL cholesterol (mg/dL)	72 (15–108)

Legend: Categorical data are presented as percentage and absolute number. Continuous data are presented as median and 95% interquartile range.

## Data Availability

The data presented in this study are available in this article and Appendix A.

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
