# Peer review of "Sarcopenia and Ghrelin System in the Clinical Outcome and Prognosis of Gastroenteropancreatic Neuroendocrine Neoplasms"

_cancers, 2021, doi:10.3390/cancers14010111_

Round 1

Reviewer 1 Report

Comments to authors:

This study demonstrated that survival is related to the nutritional status of patients with GEP-NENs and also with the molecular expression of some relevant ghrelin system components. Nutritional parameters were similar in GEP-NEN tumors of different origins. The relapsed disease is associated with lower BMI, while disease-free is associated with higher BMI at diagnosis. Patients that presented with weight loss at diagnosis had significantly lower overall survival and GOAT enzyme expression was higher in them. CT scan confirmed a high prevalence of sarcopenia and mortality occurred only in patients without sarcopenia, despite results having no statistical significance.

The findings are reasonably discussed, and support the conclusions. However, the following major points need to be addressed.

Major points

  1. This study seems to confuse the concept of Gastroenteropancreatic (GEP)-NEN. The GEP-NENs mainly fall into one of two subtypes: NETs (neuroendocrine tumors) or NECs (neuroendocrine carcinomas). Since these two types of GEP-NEN show great differences in pathology, this article would be more convincing if the authors listed the number of patients with NET and the number of patients with NEC.
  2. In the introduction, the authors mentioned that “nutritional status can be also affected by hormone secretion” (line 60-61). This study would be more clear if the authors explored the difference in nutritional status between patients with normal functions of the ghrelin system and those with abnormal functions.
  3. This article didn’t attach table1 or table S1 in either text or supplementary. We don’t know the proportion of patients with WHO grade G1, G2, or G3. We also don’t know whether there is a difference in nutritional status between patients with different grades.
  4. The line color is mistaken in Figure 2C that patients with decreased albumin should present lower OS, not a higher OS presented in Fig.2C. (The red line shall be blue, and the blue line shall be red). Figure S1B, Figure S1C, and Figure S1D seem to have the same problem.
  5. The title of this article emphasizes “CT-assessed sarcopenia” and “ghrelin system”. While CT-assessed sarcopenia only accounts for a small part of the whole text and most results in this part were not statistically significant. We would like to recommend you alter this title into a more appropriate version.
  6. In the discussion, the author claimed that a recent study reported 67% sarcopenia in 49 metastatic. But the authors didn’t appear to explain the reason why include sarcopenia as a part of this article and discuss its possible relation with GEP-NEN patients’ prognosis. The presence of this parameter (CT-assessed sarcopenia) seems abrupt.
  7. In this article, nutritional parameters, ghrelin system components, and clinical outcomes are only connected based on the statistical analysis. It would be helpful to add some demonstration about the underlying mechanisms behind them.
  8. In the discussion, the authors stated that “sarcopenia is highly prevalent in NEN patients” and we have some reservations about this statement. Under the assumption that every case in this article is well-differentiated NET, GET-NETs discussed in this article are characterized by slow proliferation and are usually asymptomatic until they enter the advanced stage or the end stage. Hence, the conclusion would be better if the authors could illustrate which stage of NEN patients is sarcopenia prevalent in.
  9. This article would be better if the authors could include a validation set.

Minor Points

  1. Skeletal muscle index, SMI in the 108th line, should be written in a full word when it is used for the first time.
  2. “One hundred four patients” in the 140th line should be corrected as “One hundred and four patients”.
  3. “any significant relation” in the 177th line should be corrected as “no significant relation”.

Author Response

REVIEWER:

This study demonstrated that survival is related to the nutritional status of patients with GEP-NENs and also with the molecular expression of some relevant ghrelin system components. Nutritional parameters were similar in GEP-NEN tumors of different origins. The relapsed disease is associated with lower BMI, while disease-free is associated with higher BMI at diagnosis. Patients that presented with weight loss at diagnosis had significantly lower overall survival and GOAT enzyme expression was higher in them. CT scan confirmed a high prevalence of sarcopenia and mortality occurred only in patients without sarcopenia, despite results having no statistical significance.

The findings are reasonably discussed, and support the conclusions. However, the following major points need to be addressed.

Major points

Reviewer: This study seems to confuse the concept of Gastroenteropancreatic (GEP)-NEN. The GEP-NENs mainly fall into one of two subtypes: NETs (neuroendocrine tumors) or NECs (neuroendocrine carcinomas). Since these two types of GEP-NEN show great differences in pathology, this article would be more convincing if the authors listed the number of patients with NET and the number of patients with NEC.

Authors: As the reviewer suggested, we specified the number of patients included in the study according to the tumor grade. This information is depicted in Table 1 of the revised version of our manuscript, and described in the results section (lines 194-195).

Reviewer: In the introduction, the authors mentioned that “nutritional status can be also affected by hormone secretion” (line 60-61). This study would be more clear if the authors explored the difference in nutritional status between patients with normal functions of the ghrelin system and those with abnormal functions.

Authors: Circulating ghrelin levels of the included patients were not available. Since this is a retrospective study of the nutritional status at diagnosis, it is not possible to evaluate this parameter.

Reviewer: This article didn’t attach table1 or table S1 in either text or supplementary. We don’t know the proportion of patients with WHO grade G1, G2, or G3. We also don’t know whether there is a difference in nutritional status between patients with different grades.

Authors: Supplemental table 1 is available in the Tables file of the revised version of our manuscript. The proportion of patients with G1, G2 or G3 are included in Table 1 of the revised version of our manuscript.

Reviewer: The line color is mistaken in Figure 2C that patients with decreased albumin should present lower OS, not a higher OS presented in Fig.2C. (The red line shall be blue, and the blue line shall be red). Figure S1B, Figure S1C, and Figure S1D seem to have the same problem.

Authors: Indeed, as the reviewer noticed, there was a mistake in the color of the lines in Figure 2C and Supp Figure 1. This fact was corrected in the revised version of our manuscript.

Reviewer: The title of this article emphasizes “CT-assessed sarcopenia” and “ghrelin system”. While CT-assessed sarcopenia only accounts for a small part of the whole text and most results in this part were not statistically significant. We would like to recommend you alter this title into a more appropriate version.

Authors: According to the suggestion of the reviewer, the title of the article was modified.

Reviewer: In the discussion, the author claimed that a recent study reported 67% sarcopenia in 49 metastatic. But the authors didn’t appear to explain the reason why include sarcopenia as a part of this article and discuss its possible relation with GEP-NEN patients’ prognosis. The presence of this parameter (CT-assessed sarcopenia) seems abrupt.

Authors: As suggested by the reviewer, the reason for evaluating sarcopenia using CT-scans in this study was detailed in the Discussion section of the revised version of our manuscript (lines 309-313).

Reviewer: In this article, nutritional parameters, ghrelin system components, and clinical outcomes are only connected based on the statistical analysis. It would be helpful to add some demonstration about the underlying mechanisms behind them.

Authors: As the reviewer suggested, in the discussion section of the revised version of our manuscript, the role of ghrelin system in tumors was deeply explained including molecular mechanisms  (lines 346-368).

Reviewer: In the discussion, the authors stated that “sarcopenia is highly prevalent in NEN patients” and we have some reservations about this statement. Under the assumption that every case in this article is well-differentiated NET, GET-NETs discussed in this article are characterized by slow proliferation and are usually asymptomatic until they enter the advanced stage or the end stage. Hence, the conclusion would be better if the authors could illustrate which stage of NEN patients is sarcopenia prevalent in.

Authors: As the reviewer suggested, the presence of sarcopenia and other nutritional parameters depending on tumor grade was explored and depicted in Supp Table 1 and reflected in the results section of the revised version of our manuscript (lines 206-208 and 248-250).

Reviewer: This article would be better if the authors could include a validation set.

Authors: We agree with the reviewer when a validation set is suggested. Since this is a unicenter study, validation of our results in a similar cohort is not possible, but we clearly reflected this point in the revised version of our manuscript (lines 385-387).

Minor Points

Reviewer: Skeletal muscle index, SMI in the 108th line, should be written in a full word when it is used for the first time.

Authors: As suggested by the reviewer, the full word of SMI was added in the revised version of our manuscript

Reviewer: “One hundred four patients” in the 140th line should be corrected as “One hundred and four patients”.

Authors: As suggested by the reviewer, this sentence was corrected.

Reviewer: “any significant relation” in the 177th line should be corrected as “no significant relation”.

Authors: As suggested by the reviewer, this sentence was corrected.

Reviewer 2 Report

The authors focused on a quite interesting topic as malnutrition can play a key role in the prognosis of GEP-NENs, however data specifically focused on this topic are scanty, thus this certainly represents a merit of current paper.

However, I'd suggest some edits to be done:

  1. Please better explain in the introduction the actual role of the ghrelin system, also based on your previous study (ref 29)
  2. Please improve the discusion by clearly eplaining the actual meaning of your finding in routine clinical practice. Would you suggest to assess the nutritional status of all GEP-NEN paptients? Please specificy the test to be requested (albumin, anthropometrici evaluation, CT etc?) at the diagnosis and during the follow-up. In the case of overt malnutrition which measures should be done? It is clear that a dietician expert in NEN should be included in the multidisciplinary management of NEN, plese clearly highlight this point.
  3. As the authors state, the main limits include the lack of a proper nutritional anthropometric evaluation and the effect of specific therapy. Regarding this last point, is it possible to add some more data re. therapies?
  4. Is there a relationship between sarcopenia/weight loss and the site of the primary tumor?
  5. minor point: please be consistent with NEN and NET throughout the whole paper. English language should be edited by a mother tongue.

Author Response

REVIEWER: The authors focused on a quite interesting topic as malnutrition can play a key role in the prognosis of GEP-NENs, however data specifically focused on this topic are scanty, thus this certainly represents a merit of current paper.

However, I'd suggest some edits to be done:

Reviewer: Please better explain in the introduction the actual role of the ghrelin system, also based on your previous study (ref 29)

Authors: As the reviewer suggested, some information about the role and expression of ghrelin system was added in the introduction and discussion of the revised version of our manuscript. (lines 101-109 and 346-368 respectively).

Reviewer: Please improve the discusion by clearly eplaining the actual meaning of your finding in routine clinical practice. Would you suggest to assess the nutritional status of all GEP-NEN paptients? Please specificy the test to be requested (albumin, anthropometrici evaluation, CT etc?) at the diagnosis and during the follow-up. In the case of overt malnutrition which measures should be done? It is clear that a dietician expert in NEN should be included in the multidisciplinary management of NEN, plese clearly highlight this point.

Authors: We agree with the reviewer concerning this point. We added some suggestions for clinical practice regarding nutrition and NENs based on the ESPEN clinical guidelines (lines 335-340).

Reviewer: As the authors state, the main limits include the lack of a proper nutritional anthropometric evaluation and the effect of specific therapy. Regarding this last point, is it possible to add some more data re. therapies?

Authors: The nutritional evaluation and the determination of ghrelin system was performed at diagnosis; thus, patients were naïve at that moment. For that reason, any specific analysis about therapies was included.

Reviewer: Is there a relationship between sarcopenia/weight loss and the site of the primary tumor?

Authors: We did not observe any relation between sarcopenia/weight loss and the primary tumor site, we clearly reflected this point in the revised version of our manuscript (lines 249-250).

Reviewer: minor point: please be consistent with NEN and NET throughout the whole paper. English language should be edited by a mother tongue.

Authors: We checked this point in the revised version of our manuscript, which was corrected by a native speaker.

Round 2

Reviewer 1 Report

Comments to authors:

The authors have revised the article according to our last comments. However, the following major points still need to be addressed. In the meantime, the text in this article should be fully reviewed and professional editing is necessary.

Major points

  1. The authors have agreed that there was a mistake in the color of the lines in Figure 2C and Supp Figure 1. However, the line color is still mistaken in Figure 2C that patients with decreased albumin should present lower OS, not a higher OS presented in Fig.2C. (The red line shall be blue, and the blue line shall be red). Supp Figure S1 also has the same problem.

Minor Points

  1. A comma should be put between two clauses in the 75th line: “in several tissues, especially in …”.
  2. The sentence in the 75th line and 76th line is fragmented, which probably should be written as: “especially in glands of the gastrointestinal tract”.
  3. “Any relation between sarcopenia …” in the 209th line should be corrected as “No relation between sarcopenia …”.

Author Response

Responses to Reviewer #1

REVIEWER:

 The authors have revised the article according to our last comments. However, the following major points still need to be addressed. In the meantime, the text in this article should be fully reviewed and professional editing is necessary.

Reviewer: The authors have agreed that there was a mistake in the color of the lines in Figure 2C and Supp Figure 1. However, the line color is still mistaken in Figure 2C that patients with decreased albumin should present lower OS, not a higher OS presented in Fig.2C. (The red line shall be blue, and the blue line shall be red). Supp Figure S1 also has the same problem.

Authors: We thank the reviewer for pointing this out. In the revised version of our manuscript Figure 2C decreased albumin is presented with the dotted line. In Supp Figure 1 the colors are inverted in comparison with figure 2.

Minor Points

Reviewer: A comma should be put between two clauses in the 75th line: “in several tissues, especially in …”.

Authors: we corrected the referred sentence.

Reviewer: The sentence in the 75th line and 76th line is fragmented, which probably should be written as: “especially in glands of the gastrointestinal tract”.

Authors: we corrected the referred sentence.

Reviewer: “Any relation between sarcopenia …” in the 209th line should be corrected as “No relation between sarcopenia …”.

Authors: we corrected the referred sentence.

Reviewer 2 Report

I'm globaòlly satisfied with the author's edits; some minor points:

  • please edit "Any relation between sarcopenia and pri-209mary tumor site was observed in our cohort" in NO relation
  • please edit specific studies reporting the expression of ghrelin system components and the nutritional relevance in NEN has not been" in specific studies...HAVE no

Author Response

Responses to Reviewer #2

Reviewer: please edit "Any relation between sarcopenia and pri-209mary tumor site was observed in our cohort" in NO relation

Authors: we corrected the referred sentence.

Reviewer: please edit specific studies reporting the expression of ghrelin system components and the nutritional relevance in NEN has not been" in specific studies...HAVE no

Authors: we corrected the referred sentence.